# Context-adaptive Entropy Model for End-to-end Optimized Image Compression

**Jooyoung Lee, Seunghyun Cho & Seung-Kwon Beack**
Broadcasting Media Research Laboratory
Electronics and Telecommunications Research Institute
Daejeon, Korea
{leejy1003,shcho,skbeack}@etri.re.kr

## Abstract

We propose a context-adaptive entropy model for use in end-to-end optimized image compression. Our model exploits two types of contexts, bit-consuming contexts and bit-free contexts, distinguished based upon whether additional bit allocation is required. Based on these contexts, we allow the model to more accurately estimate the distribution of each latent representation with a more generalized form of the approximation models, which accordingly leads to an enhanced compression performance. Based on the experimental results, the proposed method outperforms the traditional image codecs, such as BPG and JPEG2000, as well as other previous artificial-neural-network (ANN) based approaches, in terms of the peak signal-to-noise ratio (PSNR) and multi-scale structural similarity (MS-SSIM) index. The test code is publicly available at https://github.com/JooyoungLeeETRI/CA_Entropy_Model.

## 1 Introduction

Recently, artificial neural networks (ANNs) have been applied in various areas and have achieved a number of breakthroughs resulting from their superior optimization and representation learning performance. In particular, for various problems that are sufficiently straightforward that they can be solved within a short period of time by hand, a number of ANN-based studies have been conducted and significant progress has been made. With regard to image compression, however, relatively slow progress has been made owing to its complicated target problems. A number of works, focusing on the quality enhancement of reconstructed images, were proposed. For instance, certain approaches (Dong et al., 2015; Svoboda et al., 2016; Zhang et al., 2017) have been proposed to reduce artifacts caused by image compression, relying on the superior image restoration capability of an ANN. Although it is indisputable that artifact reduction is one of the most promising areas exploiting the advantages of ANNs, such approaches can be viewed as a type of post-processing, rather than image compression itself.

Regarding ANN-based image compression, the previous methods can be divided into two types. First, as a consequence of the recent success of generative models, some image compression approaches targeting the superior perceptual quality (Agustsson et al., 2018; Santurkar et al., 2018; Rippel & Bourdev, 2017) have been proposed. The basic idea here is that learning the distribution of natural images enables a very high compression level without severe perceptual loss by allowing the generation of image components, such as textures, which do not highly affect the structure or the perceptual quality of the reconstructed images. Although the generated images are very realistic, the acceptability of the machine-created image components eventually becomes somewhat application-dependent. Meanwhile, a few end-to-end optimized ANN-based approaches (Toderici et al., 2017; Johnston et al., 2018; Ballé et al., 2017; Theis et al., 2017; Ballé et al., 2018), without generative models, have been proposed. In these approaches, unlike traditional codecs comprising separate tools, such as prediction, transform, and quantization, a comprehensive solution covering all functions has been sought after using end-to-end optimization. Toderici et al. (2017)'s approach exploits a small number of latent binary representations to contain the compressed information in every step, and each step increasingly stacks the additional latent representations to achieve a progressive improvement in quality of the reconstructed images. Johnston et al. (2018) improved the compression

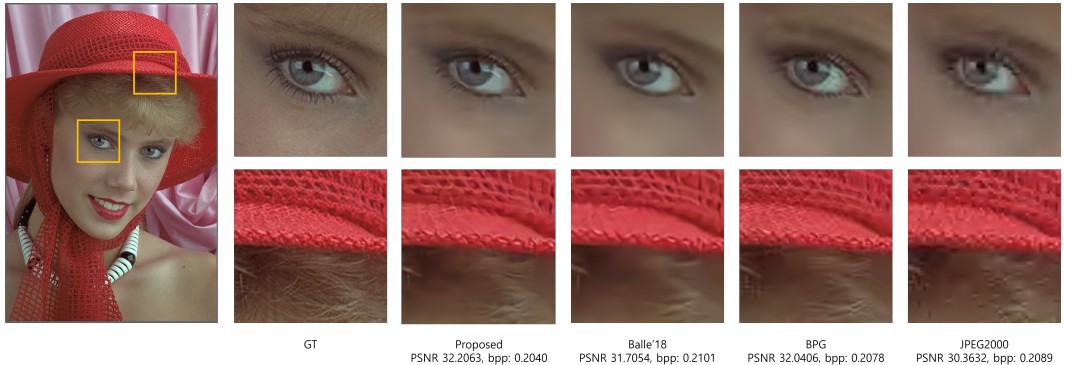

GT

Proposed
PSNR 32.2063, bpp: 0.2040

Balle'18
PSNR 31.7054, bpp: 0.2101

BPG
PSNR 32.0406, bpp: 0.2078

JPEG2000
PSNR 30.3632, bpp: 0.2089

Figure 1: Comparison of sample test results including the ground truth, our method, Ballé et al. (2018)'s approach, BPG, and JPEG2000.

performance by enhancing operation methods of the networks developed by Toderici et al. (2017). Although Toderici et al. (2017); Johnston et al. (2018) provided novel frameworks suitable to quality control using a single trained network, the increasing number of iteration steps to obtain higher image quality can be a burden to certain applications. In contrast to the approaches developed by Toderici et al. (2017) and Johnston et al. (2018), which extract binary representations with as high an entropy as possible, Ballé et al. (2017), Theis et al. (2017), and Ballé et al. (2018) regard the image compression problem as being how to retrieve discrete latent representations having as low an entropy as possible. In other words, the target problem of the former methods can be viewed as how to include as much information as possible in a fixed number of representations, whereas the latter is simply how to reduce the expected bit-rate when a sufficient number of representations are given, assuming that the low entropy corresponds to small number of bits from the entropy coder. To solve the second target problem, Ballé et al. (2017), Theis et al. (2017), and Ballé et al. (2018) adopt their own entropy models to approximate the actual distributions of the discrete latent representations. More specifically, Ballé et al. (2017) and Theis et al. (2017) proposed novel frameworks that exploit the entropy models, and proved their performance capabilities by comparing the results with those of conventional codecs such as JPEG2000. Whereas Ballé et al. (2017) and Theis et al. (2017) assume that each representation has a fixed distribution, Ballé et al. (2018) introduced an input-adaptive entropy model that estimates the scale of the distribution for each representation. This idea is based on the characteristics of natural images in which the scales of the representations vary together in adjacent areas. They provided test results that outperform all previous ANN-based approaches, and reach very close to those of BPG (Bellard, 2014), which is known as a subset of HEVC (ISO/IEC 23008-2, ITU-T H.265), used for image compression.

One of the principle elements in end-to-end optimized image compression is the trainable entropy model used for the latent representations. Because the actual distributions of latent representations are unknown, the entropy models provide the means to estimate the required bits for encoding the latent representations by approximating their distributions. When an input image $x$ is transformed into a latent representation $y$ and then uniformly quantized into $\hat{y}$, the simple entropy model can be represented by $p_{\hat{y}}(\hat{y})$, as described by Ballé et al. (2018). When the actual marginal distribution of $\hat{y}$ is denoted as $m(\hat{y})$, the rate estimation, calculated through cross entropy using the entropy model, $p_{\hat{y}}(\hat{y})$, can be represented as shown in equation (1), and can be decomposed into the actual entropy of $\hat{y}$ and the additional bits owing to a mismatch between the actual distributions and their approximations. Therefore, decreasing the rate term $R$ during the training process allows the entropy model $p_{\hat{y}}(\hat{y})$ to approximate $m(\hat{y})$ as closely as possible, and let the other parameters transform $x$ into $y$ properly such that the actual entropy of $\hat{y}$ becomes small.

$$R = \mathbb{E}_{\hat{y} \sim m}[-\log_2 p_{\hat{y}}(\hat{y})] = H(m) + D_{KL}(m||p_{\hat{y}}). \qquad (1)$$

In terms of KL-divergence, $R$ is minimized when $p_{\hat{y}}(\hat{y})$ becomes perfectly matched with the actual distribution $m(\hat{y})$. This means that the compression performance of the methods essentially depends on the capacity of the entropy model. To enhance the capacity, we propose a new entropy model

that exploits two types of contexts, bit-consuming and bit-free contexts, distinguished according to whether additional bit allocation is required. Utilizing these two contexts, we allow the model to more accurately estimate the distribution of each latent representation through the use of a more generalized form of the entropy models, and thus more effectively reduce the spatial dependencies among the adjacent latent representations. Figure 1 demonstrates a comparison of the compression results of our method to those of other previous approaches. The contributions of our work are as follows:

- We propose a new context-adaptive entropy model framework that incorporates the two different types of contexts.
- We provide the test results that outperform the widely used conventional image codec BPG in terms of the PSNR and MS-SSIM.
- We discuss the directions of improvement in the proposed methods in terms of the model capacity and the level of the contexts.

Note that we follow a number of notations given by Ballé et al. (2018) because our approach can be viewed as an extension of their work, in that we exploit the same rate-distortion (R-D) optimization framework. The rest of this paper is organized as follows. In Section 2, we introduce the key approaches of end-to-end optimized image compression and propose the context-adaptive entropy model. Section 3 demonstrates the structure of the encoder and decoder models used, and the experimental setup and results are then given in section 4. Finally, in Section 5, we discuss the current state of our work and directions for improvement.

## 2    END-TO-END OPTIMIZATION BASED ON CONTEXT-ADAPTIVE ENTROPY MODELS

### 2.1    PREVIOUS ENTROPY MODELS

Since they were first proposed by Ballé et al. (2017) and Theis et al. (2017), entropy models, which approximate the distribution of discrete latent representations, have noticeably improved the image compression performance of ANN-based approaches. Ballé et al. (2017) assumes the entropy models of the latent representations as non-parametric models, whereas Theis et al. (2017) adopted a Gaussian scale mixture model composed of six weighted zero-mean Gaussian models per representation. Although they assume different forms of entropy models, they have a common feature in that both concentrate on learning the distributions of the representations without considering input adaptivity. In other words, once the entropy models are trained, the trained model parameters for the representations are fixed for any input during the test time. Ballé et al. (2018), in contrast, introduced a novel entropy model that adaptively estimates the scales of the representations based on input. They assume that the scales of the latent representations from the natural images tend to move together within an adjacent area. To reduce this redundancy, they use a small amount of additional information by which the proper scale parameters (standard deviations) of the latent representations are estimated. In addition to the scale estimation, Ballé et al. (2018) have also shown that when the prior probability density function (PDF) for each representation in a continuous domain is convolved with a standard uniform density function, it approximates the prior probability mass function (PMF) of the discrete latent representation, which is uniformly quantized by rounding, much more closely. For training, a uniform noise is added to each latent representation so as to fit the distribution of these noisy representations into the mentioned PMF-approximating functions. Using these approaches, Ballé et al. (2018) achieved a state-of-the-art compression performance, close to that of BPG.

### 2.2    SPATIAL DEPENDENCIES OF THE LATENT VARIABLES

The latent representations, when transformed through a convolutional neural network, essentially contain spatial dependencies because the same convolutional filters are shared across the spatial regions, and natural images have various factors in common within adjacent regions. Ballé et al. (2018) successfully captured these spatial dependencies and enhanced the compression performance by input-adaptively estimating standard deviations of the latent representations. Taking a step forward, we generalize the form of the estimated distribution by allowing, in addition to the standard

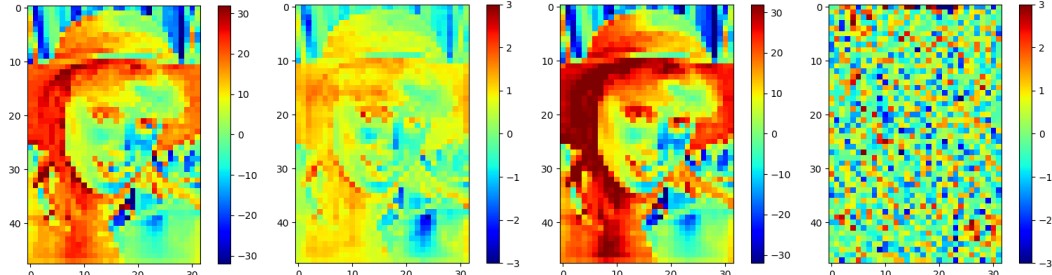

Figure 2: Examples of latent representations and their normalized versions for the two cases (the first two images show the results in which only the standard deviations are estimated using side information, whereas the last two images show the results in which the mu and standard deviation are estimated using our method). For a clear demonstration, the latent representations having the highest covariance between the spatially adjacent variables are extracted. Left: the latent representations of $\hat{y}$ from the first case. Middle left: the normalized versions of $\hat{y}$ from the first case, divided by the estimated standard deviation. Middle right: the latent variable of $\hat{y}$ from the second case. Right: the normalized versions of $\hat{y}$ from the second case, shifted and divided by the estimated mu and the standard deviation.

deviation, the mean estimation utilizing the contexts. For instance, assuming that certain representations tend to have similar values within a spatially adjacent area, when all neighborhood representations have a value of 10, we can intuitively guess that, for the current representation, the chance of having a value equal or similar to 10 is relatively high. This simple estimation will consequently reduce the entropy. Likewise, our method utilizes the given contexts for estimating the mean, as well as the standard deviation, of each latent representation. Note that Toderici et al. (2017), Johnston et al. (2018), and Rippel & Bourdev (2017) also apply context-adaptive entropy coding by estimating the probability of each binary representation. However, these context-adaptive entropy-coding methods can be viewed as separate components, rather than one end-to-end optimization component because their probability estimation does not directly contribute to the rate term of the R-D optimization framework. Figure 2 visualizes the latent variables $\hat{y}$ and their normalized versions of the two different approaches, one estimating only the standard deviation parameters and the other estimating both the mu and standard deviation parameters with the two types of mentioned contexts. The visualization shows that the spatial dependency can be removed more effectively when the mu is estimated along with the given contexts.

## 2.3 CONTEXT-ADAPTIVE ENTROPY MODEL

The optimization problem described in this paper is similar with Ballé et al. (2018), in that the input $x$ is transformed into $y$ having a low entropy, and the spatial dependencies of $y$ are captured into $\hat{z}$. Therefore, we also use four fundamental parametric transform functions: an analysis transform $g_a(x; \phi_g)$ to transform $x$ into a latent representation $y$, a synthesis transform $g_s(\hat{y}; \theta_g)$ to reconstruct image $\hat{x}$, an analysis transform $h_a(\hat{y}; \phi_h)$ to capture the spatial redundancies of $\hat{y}$ into a latent representation $z$, and a synthesis transform $h_s(\hat{z}; \theta_h)$ used to generate the contexts for the model estimation. Note that $h_s$ does not estimate the standard deviations of the representations directly as in Ballé et al. (2018)'s approach. In our method, instead, $h_s$ generates the context $c'$, one of the two types of contexts for estimating the distribution. These two types of contexts are described in this section.

Ballé et al. (2018) analyzed the optimization problem from the viewpoint of the variational autoencoder (Kingma & Welling (2014); Rezende et al. (2014)), and showed that the minimization of the KL-divergence is the same problem as the R-D optimization of image compression. Basically, we follow the same concept; however, for training, we use the discrete representations on the conditions instead of the noisy representations, and thus the noisy representations are only used as the inputs to the entropy models. Empirically, we found that using discrete representations on the conditions show better results, as shown in appendix 6.2. These results might come from removing the mis-

matches of the conditions between the training and testing, thereby enhancing the training capacity by limiting the affect of the uniform noise only to help the approximation to the probability mass functions. We use the gradient overriding method with the identity function, as in Theis et al. (2017), to deal with the discontinuities from the uniform quantization. The resulting objective function used in this paper is given in equation (2). The total loss consists of two terms representing the rates and distortions, and the coefficient $\lambda$ controls the balance between the rate and distortion during the R-D optimization. Note that $\lambda$ is not an optimization target, but a manually configured condition that determines which to focus on between rate and distortion:

$$\mathcal{L} = R + \lambda D \tag{2}$$
$$\text{with } R = \mathbb{E}_{\boldsymbol{x} \sim p_{\boldsymbol{x}}} \mathbb{E}_{\tilde{\boldsymbol{y}}, \tilde{\boldsymbol{z}} \sim q} \Big[ -\log p_{\tilde{\boldsymbol{y}}|\hat{\boldsymbol{z}}}(\tilde{\boldsymbol{y}} \mid \hat{\boldsymbol{z}}) - \log p_{\tilde{\boldsymbol{z}}}(\tilde{\boldsymbol{z}}) \Big],$$
$$D = \mathbb{E}_{\boldsymbol{x} \sim p_{\boldsymbol{x}}} \Big[ -\log p_{\boldsymbol{x}|\hat{\boldsymbol{y}}}(\boldsymbol{x} \mid \hat{\boldsymbol{y}}) \Big]$$

Here, the noisy representations of $\tilde{\boldsymbol{y}}$ and $\tilde{\boldsymbol{z}}$ follow the standard uniform distribution, the mean values of which are $\boldsymbol{y}$ and $\boldsymbol{z}$, respectively, when $\boldsymbol{y}$ and $\boldsymbol{z}$ are the result of the transforms $g_a$ and $h_a$, repectively. Note that the input to $h_a$ is $\hat{\boldsymbol{y}}$, which is a uniformly quantized representation of $\boldsymbol{y}$, rather than the noisy representation $\tilde{\boldsymbol{y}}$. $Q$ denotes the uniform quantization function, for which we simply use a rounding function:

$$q(\tilde{\boldsymbol{y}}, \tilde{\boldsymbol{z}} \mid \boldsymbol{x}, \boldsymbol{\phi}_g, \boldsymbol{\phi}_h) = \prod_i \mathcal{U}\big(\tilde{y}_i \mid y_i - \tfrac{1}{2}, y_i + \tfrac{1}{2}\big) \cdot \prod_j \mathcal{U}\big(\tilde{z}_j \mid z_j - \tfrac{1}{2}, z_j + \tfrac{1}{2}\big) \tag{3}$$
$$\text{with } \boldsymbol{y} = g_a(\boldsymbol{x}; \boldsymbol{\phi}_g), \hat{\boldsymbol{y}} = Q(\boldsymbol{y}), \boldsymbol{z} = h_a(\hat{\boldsymbol{y}}; \boldsymbol{\phi}_h).$$

The rate term represents the expected bits calculated using the entropy models of $p_{\tilde{\boldsymbol{y}}|\hat{\boldsymbol{z}}}$ and $p_{\tilde{\boldsymbol{z}}}$. Note that $p_{\tilde{\boldsymbol{y}}|\hat{\boldsymbol{z}}}$ and $p_{\tilde{\boldsymbol{z}}}$ are eventually the approximations of $p_{\hat{\boldsymbol{y}}|\hat{\boldsymbol{z}}}$ and $p_{\hat{\boldsymbol{z}}}$, respectively. Equation (4) represents the entropy model for approximating the required bits for $\hat{\boldsymbol{y}}$. The model is based on the Gaussian model, which not only has the standard deviation parameter $\sigma_i$, but also the mu parameter $\mu_i$. The values of $\mu_i$ and $\sigma_i$ are estimated from the two types of given contexts based on the function $f$, the distribution estimator, in a deterministic manner. The two types of contexts, bit-consuming and bit-free contexts, for estimating the distribution of a certain representation are denoted as $\boldsymbol{c}_i'$ and $\boldsymbol{c}_i''$. $E'$ extracts $\boldsymbol{c}_i'$ from $\boldsymbol{c}'$, the result of transform $h_s$. In contrast to $\boldsymbol{c}_i'$, no additional bit allocation is required for $\boldsymbol{c}_i''$. Instead, we simply utilize the known (already entropy-coded or decoded) subset of $\hat{\boldsymbol{y}}$, denoted as $\langle \hat{\boldsymbol{y}} \rangle$. Here, $\boldsymbol{c}_i''$ is extracted from $\langle \hat{\boldsymbol{y}} \rangle$ by the extractor $E''$. We assume that the entropy coder and the decoder sequentially process $\hat{y}_i$ in the same specific order, such as with raster scanning, and thus $\langle \hat{\boldsymbol{y}} \rangle$ given to the encoder and decoder can always be identical when processing the same $\hat{y}_i$. A formal expression of this is as follows:

$$p_{\tilde{\boldsymbol{y}}|\hat{\boldsymbol{z}}}(\tilde{\boldsymbol{y}} \mid \hat{\boldsymbol{z}}, \boldsymbol{\theta}_h) = \prod_i \Big( \mathcal{N}\big(\mu_i, \sigma_i^2\big) * \mathcal{U}\big(-\tfrac{1}{2}, \tfrac{1}{2}\big) \Big)(\tilde{y}_i) \tag{4}$$
$$\text{with } \mu_i, \sigma_i = f(\boldsymbol{c}_i', \boldsymbol{c}_i''),$$
$$\boldsymbol{c}_i' = E'(h_s(\hat{\boldsymbol{z}}; \boldsymbol{\theta}_h), i),$$
$$\boldsymbol{c}_i'' = E''(\langle \hat{\boldsymbol{y}} \rangle, i),$$
$$\hat{\boldsymbol{z}} = Q(\boldsymbol{z})$$

In the case of $\hat{\boldsymbol{z}}$, a simple entropy model is used. We assumed that the model follows zero-mean Gaussian distributions which have a trainable $\boldsymbol{\sigma}$. Note that $\hat{\boldsymbol{z}}$ is regarded as side information and it contributes a very small amount of the total bit-rate, as described by Ballé et al. (2018), and thus we use this simpler version of the entropy model, rather than a more complex model, for end-to-end optimization over all parameters of the proposed method:

$$p_{\hat{\boldsymbol{z}}}(\tilde{\boldsymbol{z}}) = \prod_j \Big( \mathcal{N}\big(0, \sigma_j^2\big) * \mathcal{U}\big(-\tfrac{1}{2}, \tfrac{1}{2}\big) \Big)(\tilde{z}_j) \tag{5}$$

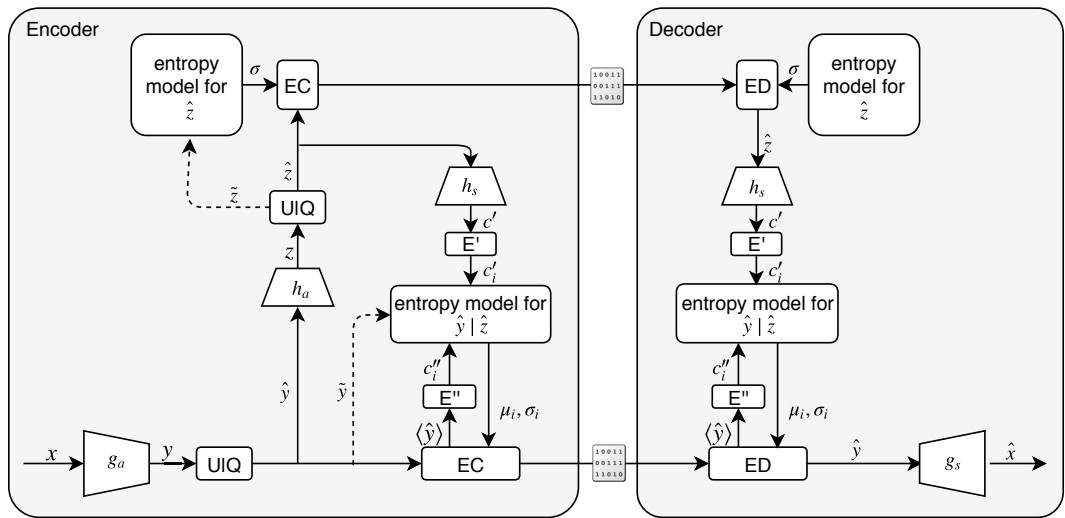

Figure 3: Encoder-decoder model of the proposed method. The left block represents the encoder side, whereas the right block represents the deocder side. The small icons between them represent the entropy-coded bitstreams. EC and ED represent entropy coding and entropy decoding, and U | Q represents the addition of uniform noise to $y$ or a uniform quantization of $y$. Noisy representations are used only for training as inputs to the entropy models, and are illustrated with the dotted lines.

Note that actual entropy coding or decoding processes are not necessarily required for training or encoding because the rate term is not the amount of real bits, but an estimation calculated from the entropy models, as mentioned previously. We calculate the distortion term using the mean squared error (MSE)[1], assuming that $p_{x|\hat{y}}$ follows a Gaussian distribution as a widely used distortion metric.

## 3 ENCODER-DECODER MODEL

This section describes the basic structure of the proposed encoder-decoder model. On the encoder side, an input image is transformed into latent representations, quantized, and then entropy-coded using the trained entropy models. In contrast, the decoder first applies entropy decoding with the same entropy models used for the encoder, and reconstructs the image from the latent representations, as illustrated in figure 3. It is assumed that all parameters that appear in this section were already trained. The structure of the encoder-decoder model basically includes $g_a$ and $g_s$ in charge of the transform of $x$ into $y$ and its inverse transform, respectively. The transformed $y$ is uniformly quantized into $\hat{y}$ by rounding. Note that, in the case of approaches based on the entropy models, unlike traditional codecs, tuning the quantization steps is usually unnecessary because the scales of the representations are optimized together through training. The other components between $g_a$ and $g_s$ carry out the role of entropy coding (or decoding) with the shared entropy models and underlying context preparation processes. More specifically, the entropy model estimates the distribution of each $\hat{y}_i$ individually, in which $\mu_i$ and $\sigma_i$ are estimated with the two types of given contexts, $c'_i$ and $c''_i$. Among these contexts, $c'$ can be viewed as side information, which requires an additional bit allocation. To reduce the required bit-rate for carrying $c'$, the latent representation $z$, transformed from $\hat{y}$, is quantized and entropy-coded by its own entropy model, as specified in section 2.3. On the other hand, $c''_i$ is extracted from $\langle \hat{y} \rangle$, without any additional bit allocation. Note that $\langle \hat{y} \rangle$ varies as the entropy coding or decoding progresses, but is always identical for processing the same $\hat{y}_i$ in both the encoder and decoder, as described in 2.3. The parameters of $h_s$ and the entropy models are simply shared by both the encoder and the decoder. Note that the inputs to the entropy models during training are the noisy representations, as illustrated with the dotted line in figure 3, to allow the entropy model to approximate the probability mass functions of the discrete representations.

---

[1]We also provide supplemental experiment results in which an MS-SSIM (Wang et al. (2003)) based distortion term is used for optimization. To calculate the distortion term, we multiplied 1-MS-SSIM by 3000.

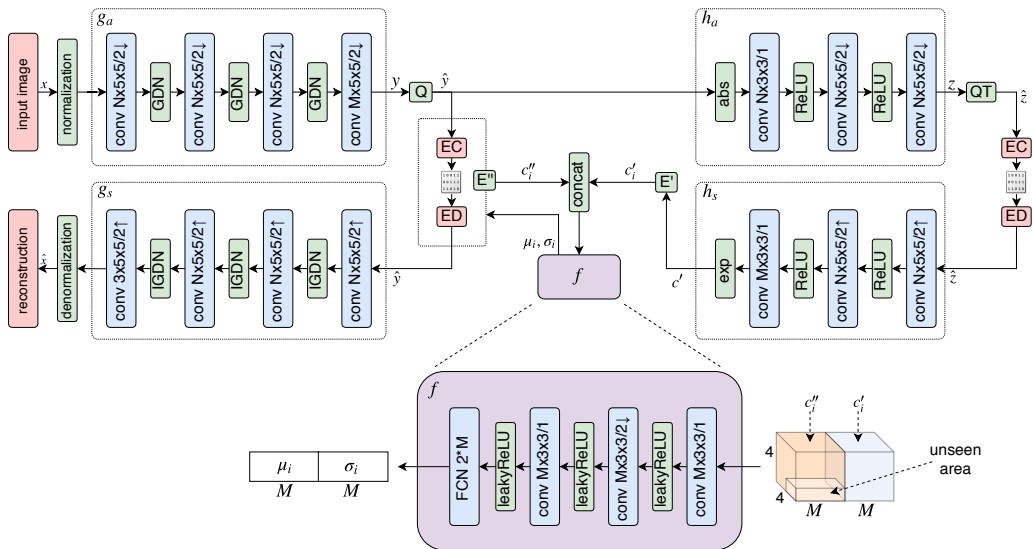

Figure 4: Implementation of the proposed method. We basically use the convolutional autoencoder structure, and the distribution estimator $f$ is also implemented using convolutional neural networks. The notations of the convolutional layer follow Ballé et al. (2018): the number of filters $\times$ filter height $\times$ filter width / the downscale or upscale factor, where $\uparrow$ and $\downarrow$ denote the up and downscaling, respectively. For up or downscaling, we use the transposed convolution. For the networks, input images are normalized into a scale between -1 and 1.

## 4 EXPERIMENTS

### 4.1 IMPLEMENTATION

We use a convolutional neural networks to implement the analysis transform and the synthesis transform functions, $g_a$, $g_s$, $h_a$, and $h_s$. The structures of the implemented networks follow the same structures of Ballé et al. (2018), except that we use the exponentiation operator instead of an absolute operator at the end of $h_s$. Based on Ballé et al. (2018)'s structure, we added the components to estimate the distribution of each $\hat{y}_i$, as shown in figure 4. Herein, we represent a uniform quantization (round) as "Q," entropy coding as "EC," and entropy decoding as "ED." The distribution estimator is denoted as $f$, and is also implemented using the convolutional layers which takes channel-wise concatenated $c_i'$ and $c_i''$ as inputs and provides estimated $\mu_i$ and $\sigma_i$ as results. Note that the same $c_i'$ and $c_i''$ are shared for all $\hat{y}_i$s located at the same spatial position. In other words, we let $E'$ extract all spatially adjacent elements from $c'$ across the channels to retrieve $c_i'$ and likewise let $E''$ extract all adjacent known elements from $\langle \hat{y} \rangle$ for $c_i''$. This could have the effect of capturing the remaining correlations among the different channels. In short, when $M$ is the total number of channels of $y$, we let $f$ estimate all $M$ distributions of $\hat{y}_i$s, which are located at the same spatial position, using only a single step, thereby allowing the total number of estimations to be reduced. Furthermore, the parameters of $f$ are shared for all spatial positions of $\hat{y}$, and thus only one trained $f$ per $\lambda$ is necessary to process any sized images. In the case of training, however, collecting the results from the all spatial positions to calculate the rate term becomes a significant burden, despite the simplifications mentioned above. To reduce this burden, we designate a certain number (32 and 16 for the base model and the hybrid model, respectively) of random spatial points as the representatives per training step, to calculate the rate term easily. Note that we let these random points contribute solely to the rate term, whereas the distortion is still calculated over all of the images.

Because $y$ is a three-dimensional array in our implementation, index $i$ can be represented as three indexes, $k$, $l$, and $m$, representing the horizontal index, the vertical index, and the channel index, respectively. When the current position is given as $(k, l, m)$, $E'$ extracts $c'_{[k-2...k+1],[l-3...l],[1...M]}$ as $c_i'$, and $E''$ extracts $\langle \hat{y} \rangle_{[k-2...k+1],[l-3...l],[1...M]}$ as $c_i''$, when $\langle \hat{y} \rangle$ represents the known area of $\hat{y}$. Note that we filled in the unknown area of $\langle \hat{y} \rangle$ with zeros, to maintain the dimensions of $\langle \hat{y} \rangle$

identical to those of $\hat{y}$. Consequently, $c''_{i\ [3...4],4,[1...M]}$ are always padded with zeros. To keep the dimensions of the estimation results to the inputs, the marginal areas of $c'$ and $\langle\hat{y}\rangle$ are also set to zeros. Note that when training or encoding, $c''_i$ can be extracted using simple $4{\times}4{\times}M$ windows and binary masks, thereby enabling parallel processing, whereas a sequential reconstruction is inevitable for decoding.

Another implementation technique used to reduce the implementation cost is combining the lightweight entropy model with the proposed model. The lightweight entropy model assumes that the representations follow a zero-mean Gaussian model with the estimated standard deviations, which is very similar with Ballé et al. (2018)'s approach. We utilize this hybrid approach for the top four cases, in bit-rate descending order, of the nine $\lambda$ configurations, based on the assumption that for the higher-quality compression, the number of sparse representations having a very low spatial dependency increases, and thus a direct scale estimation provides sufficient performance for these added representations. For implementation, we separate the latent representation $y$ into two parts, $y_1$ and $y_2$, and two different entropy models are applied for them. Note that the parameters of $g_a$, $g_s$, $h_a$, and $h_s$ are shared, and all parameters are still trained together. The detailed structure and experimental settings are described in appendix 6.1.

The number of parameters $N$ and $M$ are set to 128 and 192, respectively, for the five $\lambda$ configurations for lower bit-rates, whereas 2-3 times more parameters, described in appendix 6.1, are used for the four $\lambda$ configurations for higher bit-rates. Tensorflow and Python were used to setup the overall network structures, and for the actual entropy coding and decoding using the estimated model parameters, we implemented an arithmetic coder and decoder, for which the source code of the "Reference arithmetic coding" project[2] was used as the base code.

## 4.2 EXPERIMENTAL ENVIRONMENTS

We optimized the networks using two different types of distortion terms, one with MSE and the other with MS-SSIM. For each distortion type, the average bits per pixel (BPP) and the distortion, PSNR and MS-SSIM, over the test set are measured for each of the nine $\lambda$ configurations. Therefore, a total of 18 networks are trained and evaluated within the experimental environments, as explained below:

- For training, we used $256{\times}256$ patches extracted from 32,420 randomly selected YFCC100m (Thomee et al. (2016)) images. We extracted one patch per image, and the extracted regions were randomly chosen. Each batch consists of eight images, and 1M iterations of the training steps were conducted, applying the ADAM optimizer (Kingma & Ba (2015)). We set the initial learning rate to $5{\times}10^-5$, and reduced the rate by half every 50,000 iterations for the last 200,000 iterations. Note that, in the case of the four $\lambda$ configurations for high bpp, in which the hybrid entropy model is used, 1M iterations of pre-training steps were conducted using the learning rate of $1{\times}10^-5$. Although we previously indicated that the total loss is the sum of $R$ and $\lambda D$ for a simple explanation, we tuned the balancing parameter $\lambda$ in a similar way as Theis et al. (2017), as indicated in equation (6). We used the $\lambda$ parameters ranging from 0.01 to 0.5.

$$\mathcal{L} = \frac{\lambda}{W_y \cdot H_y \cdot 256}R + \frac{1-\lambda}{1000}D. \tag{6}$$

- For the evaluation, we measured the average BPP and average quality of the reconstructed images in terms of the PSNR and MS-SSIM over 24 PNG images of the Kodak PhotoCD image dataset (Kodak, 1993). Note that we represent the MS-SSIM results in the form of decibels, as in Ballé et al. (2018), to increase the discrimination.

## 4.3 EXPERIMENTAL RESULTS

We compared the test results with other previous methods, including traditional codecs such as BPG and JPEG2000, as well as previous ANN-based approaches such as Theis et al. (2017) and Ballé

---

[2]Nayuki, "Reference arithmetic coding," https://github.com/nayuki/Reference-arithmetic-coding, Copyright © 2018 Project Nayuki. (MIT License).

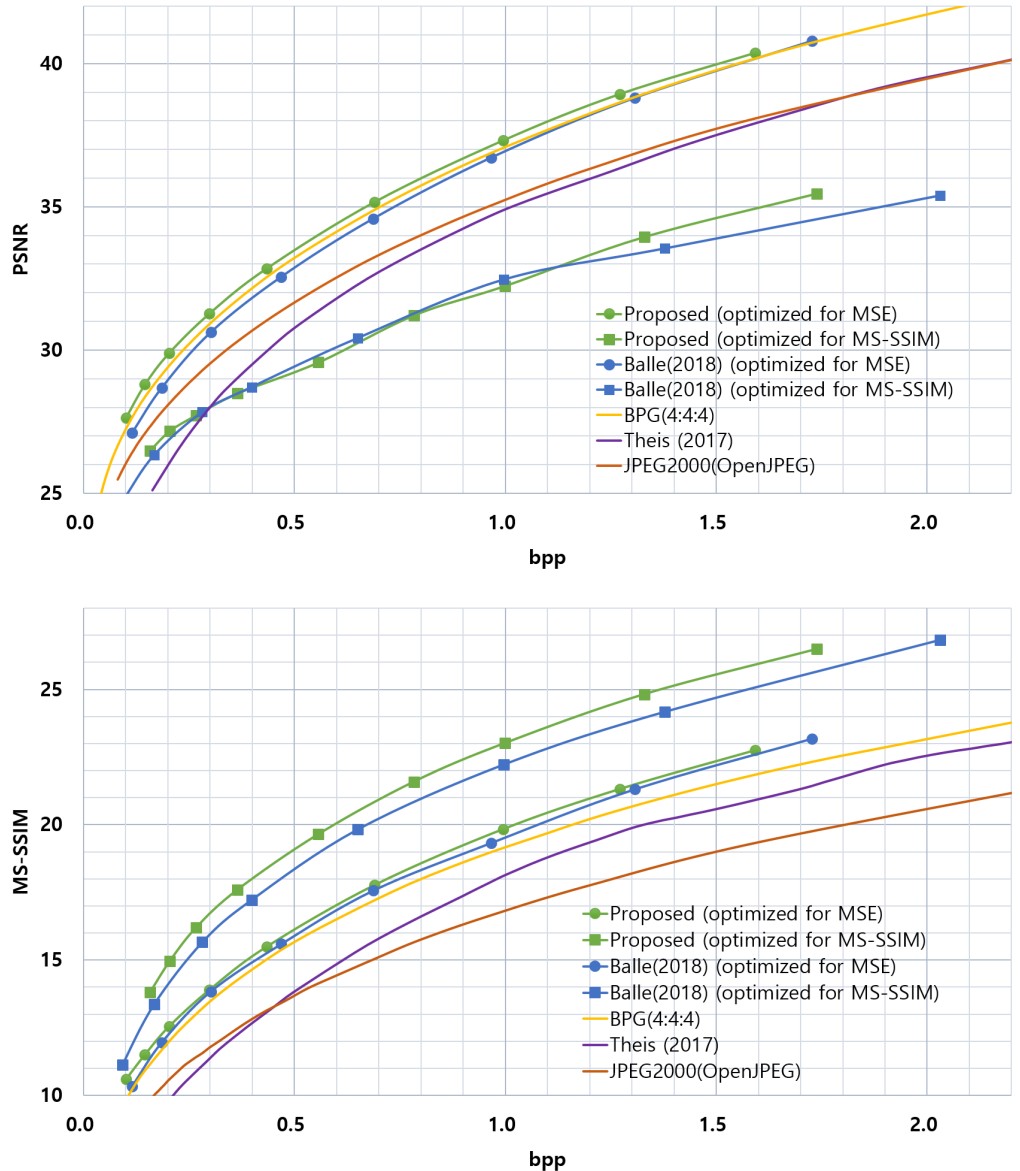

Figure 5: Rate–distortion curves of the proposed method and competitive methods. The top plot represents the PSNR values as a result of changes in bpp, whereas the bottom plot shows MS-SSIM values in the same manner. Note that MS-SSIM values are converted into decibels($-10 \log_{10}(1 - \text{MS-SSIM})$) for differentiating the quality levels, in the same manner as in Ballé et al. (2018).

et al. (2018). Because two different quality metrics are used, the results are presented with two separate plots. As shown in figure 5, our methods outperform all other previous methods in both metrics. In particular, our models not only outperform Ballé et al. (2018)'s method, which is believed to be a state-of-the-art ANN-based approach, but we also obtain better results than the widely used conventional image codec, BPG.

More specifically, the compression gains in terms of the BD-rate of PSNR over JPEG2000, Ballé et al. (2018)'s approach (MSE-optimized), and BPG are 34.08%, 11.97%, and 6.85%, respectively. In the case of MS-SSIM, we found wider gaps of 68.82%, 13.93%, and 49.68%, respectively. Note that we achieved significant gains over traditional codecs in terms of MS-SSIM, although this might be because the dominant target metric of the traditional codec developments have been the PSNR.

In other words, they can be viewed as a type of MSE-optimized codec. Even when setting aside the case of MS-SSIM, our results can be viewed as one concrete evidence supporting that ANN-based image compression can outperform the existing traditional image codecs in terms of the compression performance. Supplemental image samples are provided in appendix 6.3.

## 5 DISCUSSION

Based on previous ANN-based image compression approaches utilizing entropy models (Ballé et al., 2017; Theis et al., 2017; Ballé et al., 2018), we extended the entropy model to exploit two different types of contexts. These contexts allow the entropy models to more accurately estimate the distribution of the representations with a more generalized form having both mean and standard deviation parameters. Based on the evaluation results, we showed the superiority of the proposed method. The contexts we utilized are divided into two types. One is a sort of free context, containing the part of the latent variables known to both the encoder and the decoder, whereas the other is the context, which requires additional bit allocation. Because the former is a generally used context in a variety of codecs, and the latter was already verified to help compression using Ballé et al. (2018)'s approach, our contributions are not the contexts themselves, but can be viewed as providing a framework of entropy models utilizing these contexts.

Although the experiments showed the best results in the ANN-based image compression domain, we still have various studies to conduct to further improve the performance. One possible way is generalizing the distribution models underlying the entropy model. Although we enhanced the performance by generalizing the previous entropy models, and have achieved quite acceptable results, the Gaussian-based entropy models apparently have a limited expression power. If more elaborate models, such as the non-parametric models of Ballé et al. (2018) or Oord et al. (2016), are combined with the context-adaptivity proposed in this paper, they would provide better results by reducing the mismatch between the actual distributions and the approximation models. Another possible way is improving the level of the contexts. Currently, our methods only use low-level representations within very limited adjacent areas. However, if the sufficient capacity of the networks and higher-level contexts are given, a much more accurate estimation could be possible. For instance, if an entropy model understands the structures of human faces, in that they usually have two eyes, between which a symmetry exists, the entropy model could approximate the distributions more accurately when encoding the second eye of a human face by referencing the shape and position of the first given eye. As is widely known, various generative models (Goodfellow et al., 2014; Radford et al., 2016; Zhao et al., 2017) learn the distribution $p(\boldsymbol{x})$ of the images within a specific domain, such as human faces or bedrooms. In addition, various in-painting methods (Pathak et al., 2016; Yang et al., 2017; Yeh et al., 2017) learn the conditional distribution $p(\boldsymbol{x} \mid context)$ when the viewed areas are given as $context$. Although these methods have not been developed for image compression, hopefully such high-level understandings can be utilized sooner or later. Furthermore, the contexts carried using side information can also be extended to some high-level information such as segmentation maps or any other information that helps with compression. Segmentation maps, for instance, may be able to help the entropy models estimate the distribution of a representation discriminatively according to the segment class the representation belongs to.

Traditional codecs have a long development history, and a vast number of hand-crafted heuristics have been stacked thus far, not only for enhancing compression performance, but also for compromising computational complexities. Therefore, ANN-based image compression approaches may not provide satisfactory solutions as of yet, when taking their high complexity into account. However, considering its much shorter history, we believe that ANN-based image compression has much more potential and possibility in terms of future extension. Although we remain a long way from completion, we hope the proposed context-adaptive entropy model will provide an useful contribution to this area.

### ACKNOWLEDGMENTS

This work was supported by Institute for Information & communications Technology Promotion(IITP) grant funded by the Korea government (MSIT) (No. 2017-0-00072, Development of Audio/Video Coding and Light Field Media Fundamental Technologies for Ultra Realistic Teramedia).

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

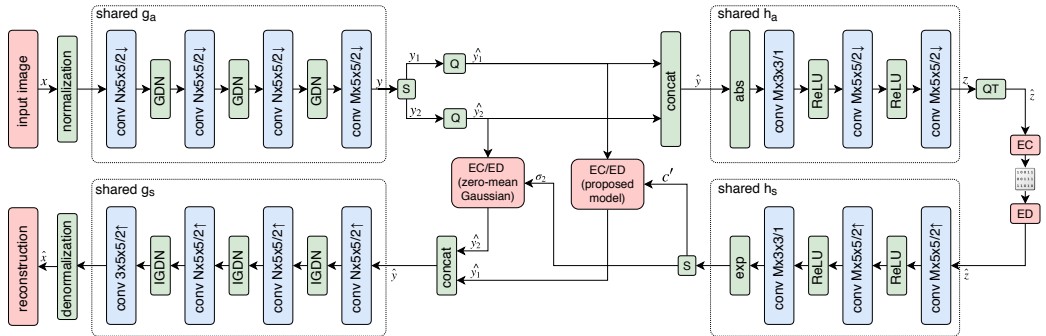

Figure 6: The structure of the hybrid network for higher bit-rate environments. The same notations as in the figure 4 are used. The representation $\boldsymbol{y}$ is divided into two parts and quantized. One of the resulting parts, $\hat{\boldsymbol{y_1}}$, is encoded using the proposed model, whereas the other, $\hat{\boldsymbol{y_2}}$, is encoded using a simpler model in which only the standard deviations are estimated using side information. The detailed structure of the proposed model is illustrated in figure 4. All concatenation and split operations are performed in a channel-wise manner.

# 6 APPENDIX

## 6.1 HYBRID NETWORK FOR HIGHER BIT-RATE COMPRESSIONS

We combined the lightweight entropy model with the context-adaptive entropy model to reduce the implementation costs for high-bpp configurations. The lightweight model exploits the scale (standard deviation) estimation, assuming that the PMF approximations of the quantized representations follow zero-mean Gaussian distributions convolved with a standard uniform distribution.

Figure 6 illustrates the network structure of this hybrid network. The representation $\boldsymbol{y}$ is split channel-wise into two parts, $\boldsymbol{y_1}$ and $\boldsymbol{y_2}$, which have $M_1$ and $M_2$ channels, respectively, and is then quantized. Here, $\hat{\boldsymbol{y_1}}$ is entropy coded using the proposed entropy model, whereas $\hat{\boldsymbol{y_2}}$ is coded with the lightweight entropy model. The standard deviations of $\hat{\boldsymbol{y_2}}$ are estimated using $h_a$ and $h_s$. Unlike the context-adaptive entropy model, which uses the results of $h_a$ ($\hat{\boldsymbol{c'}}$) as the input source to the estimator $f$, the lightweight entropy model retrieves the estimated standard deviations from $h_a$ directly. Note that $h_a$ takes the concatenated $\hat{\boldsymbol{y_1}}$ and $\hat{\boldsymbol{y_2}}$ as input, and $h_s$ generates $\hat{\boldsymbol{c'}}$ as well as $\boldsymbol{\sigma}_2$, at the same time.

The total loss function also consists of the rate and distortion terms, although the rate is divided into three parts, each of which is for $\hat{\boldsymbol{y_1}}$, $\hat{\boldsymbol{y_2}}$, and $\hat{\boldsymbol{z}}$, respectively. The distortion term is the same as before, but note that $\hat{\boldsymbol{y}}$ is the channel-wise concatenated representation of $\hat{\boldsymbol{y_1}}$ and $\hat{\boldsymbol{y_2}}$:

$$\mathcal{L} = R + \lambda D \tag{7}$$
$$\text{with } R = \mathbb{E}_{\boldsymbol{x} \sim p_{\boldsymbol{x}}} \mathbb{E}_{\tilde{\boldsymbol{y_1}}, \tilde{\boldsymbol{y_2}}, \tilde{\boldsymbol{z}} \sim q} \Big[ -\log p_{\tilde{\boldsymbol{y_1}}|\hat{\boldsymbol{z}}}(\tilde{\boldsymbol{y_1}} \mid \hat{\boldsymbol{z}}) - \log p_{\tilde{\boldsymbol{y_2}}|\hat{\boldsymbol{z}}}(\tilde{\boldsymbol{y_2}} \mid \hat{\boldsymbol{z}}) - \log p_{\tilde{\boldsymbol{z}}}(\tilde{\boldsymbol{z}}) \Big],$$
$$D = \mathbb{E}_{\boldsymbol{x} \sim p_{\boldsymbol{x}}} \Big[ -\log p_{\boldsymbol{x}|\hat{\boldsymbol{y}}}(\boldsymbol{x} \mid \hat{\boldsymbol{y}}) \Big]$$

Here, the noisy representations of $\tilde{\boldsymbol{y_1}}$, $\tilde{\boldsymbol{y_2}}$, and $\tilde{\boldsymbol{z}}$ follow a standard uniform distribution, the mean values of which are $\boldsymbol{y_1}$, $\boldsymbol{y_2}$, and $\boldsymbol{z}$, respectively. In addition, $\boldsymbol{y_1}$ and $\boldsymbol{y_2}$ are channel-wise split representations from $\boldsymbol{y}$, the results of the transform $g_a$, and have $M_1$ and $M_2$ channels, respectively:

$$
\begin{aligned}
q(\tilde{\boldsymbol{y}}_1, \tilde{\boldsymbol{y}}_2, \tilde{\boldsymbol{z}} \mid \boldsymbol{x}, \boldsymbol{\phi}_g, \boldsymbol{\phi}_h) \quad = \quad & \prod_i \mathcal{U}\left(\tilde{y}_{1i} \mid y_{1i} - \tfrac{1}{2}, y_{1i} + \tfrac{1}{2}\right) \cdot \\
& \prod_j \mathcal{U}\left(\tilde{y}_{2j} \mid y_{2j} - \tfrac{1}{2}, y_{2j} + \tfrac{1}{2}\right) \cdot \\
& \prod_k \mathcal{U}\left(\tilde{z}_k \mid z_k - \tfrac{1}{2}, z_k + \tfrac{1}{2}\right) \quad\quad (8)
\end{aligned}
$$

$$
\text{with } \boldsymbol{y_1}, \boldsymbol{y_2} = S(g_a(\boldsymbol{x}; \boldsymbol{\phi}_g)), \hat{\boldsymbol{y}} = Q(\boldsymbol{y}_1) \oplus Q(\boldsymbol{y}_2), \boldsymbol{z} = h_a(\hat{\boldsymbol{y}}; \boldsymbol{\phi}_h).
$$

The rate term for $\hat{\boldsymbol{y}}_1$ is the same model as that of equation (4). Note that $\hat{\boldsymbol{\sigma}_2}$ does not contribute here, but does contribute to the model for $\hat{\boldsymbol{y_2}}$:

$$
p_{\tilde{\boldsymbol{y_1}} \mid \hat{\boldsymbol{z}}}(\tilde{\boldsymbol{y_1}} \mid \hat{\boldsymbol{z}}, \boldsymbol{\theta}_h) = \prod_i \left( \mathcal{N}\left(\mu_{1i}, \sigma_{1i}{}^2\right) * \mathcal{U}\left(-\tfrac{1}{2}, \tfrac{1}{2}\right) \right)(\tilde{y}_{1i}) \quad\quad (9)
$$

$$
\begin{aligned}
\text{with } \mu_{1i}, \sigma_{1i} &= f(\boldsymbol{c}_i', \boldsymbol{c}_i''), \\
\boldsymbol{c}_i' &= E'(\boldsymbol{c}', i), \\
\boldsymbol{c}_i'' &= E''(\langle \hat{\boldsymbol{y_1}} \rangle, i), \\
\boldsymbol{c}', \boldsymbol{\sigma_2} &= S(h_s(\hat{\boldsymbol{z}}; \boldsymbol{\theta}_h))
\end{aligned}
$$

The rate term for $\hat{\boldsymbol{y_2}}$ is almost the same as Ballé et al. (2018), except that noisy representations are only used as the inputs to the entropy models for training, and not for the conditions of the models.

$$
p_{\tilde{\boldsymbol{y_2}} \mid \hat{\boldsymbol{z}}}(\tilde{\boldsymbol{y_2}} \mid \hat{\boldsymbol{z}}, \boldsymbol{\theta}_h) \quad = \quad \prod_j \left( \mathcal{N}\left(0, \sigma_{2j}{}^2\right) * \mathcal{U}\left(-\tfrac{1}{2}, \tfrac{1}{2}\right) \right)(\tilde{y}_{2j}) \quad\quad (10)
$$

The model of $\boldsymbol{z}$ is the same as in equation (5). For implementation, we used this hybrid structure for the top-four configurations in bit-rate descending order. We set $N$, $M_1$, and $M_2$ to 400, 192, and 408, respectively, for the top-two configurations, and to 320, 192, and 228, respectively, for the next two configurations.

In addition, we measured average execution time per image, spent for encoding and decoding Kodak PhotoCD image dataset (Kodak, 1993), to clarify benefit of the hybrid model. The test was conducted under CPU environments, Intel i9-7900X. Note that we ignored time for actual entropy coding because all models with the same values of $N$ and $M$ spend the same amount of time for entropy coding. As shown in figure 7, the hybrid models clearly reduced execution time of the models. Setting $N$ and $M$ to 320 and 420, respectively, we obtained 46.83% of speed gain. With the higher number of parameters, 400 of $N$ and 600 of $M$, we obtained 57.28% of speed gain.

## 6.2 TEST RESULTS OF THE MODELS TRAINED USING DIFFERENT TYPES OF REPRESENTATIONS

In this section, we provide test results of the two models, the proposed model trained using discrete representations as inputs to the synthesis transforms, $g_s$ and $h_s$, and the same model but trained using noisy representations following the training process of Ballé et al. (2018)'s approach. In detail, in training phase of the proposed model, we used quantized representations $\hat{\boldsymbol{y}}$ and $\hat{\boldsymbol{z}}$ as inputs to the transforms $g_s$ and $h_s$, respectively, to ensure the same conditions of training and testing phases. On the other hand, for training the compared model, representations $\tilde{\boldsymbol{y}}$ and $\tilde{\boldsymbol{z}}$ are used as inputs to the transforms. An additional change of the proposed model is using $\hat{\boldsymbol{y}}$, instead of $\boldsymbol{y}$, as inputs to $h_a$, but note that this has nothing to do with the mismatches between training and testing. We used them to match inputs to $h_a$ to targets of model estimation via $f$. As shown in figure 8, the proposed model, trained using discrete representations, was 5.94% better than the model trained using noisy representations, in terms of the BD-rate of PSNR. Compared with Ballé et al. (2018)'s approach, the performance gains of the two models, trained using discrete representations and noisy representations, were 11.97% and 7.20%, respectively.

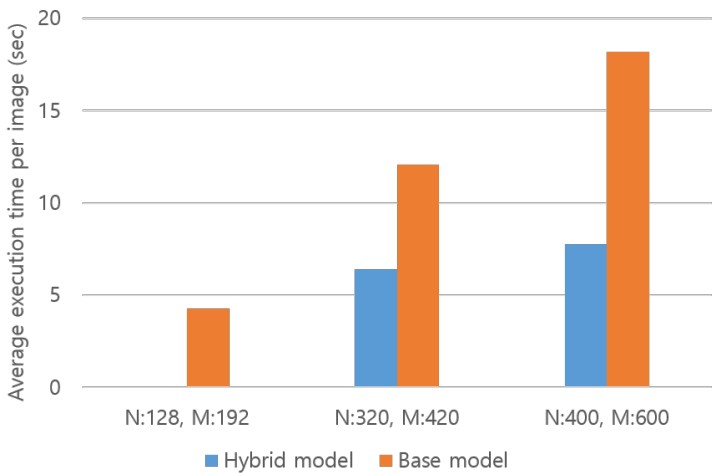

Figure 7: Comparison results of execution time between the base model and hybrid model

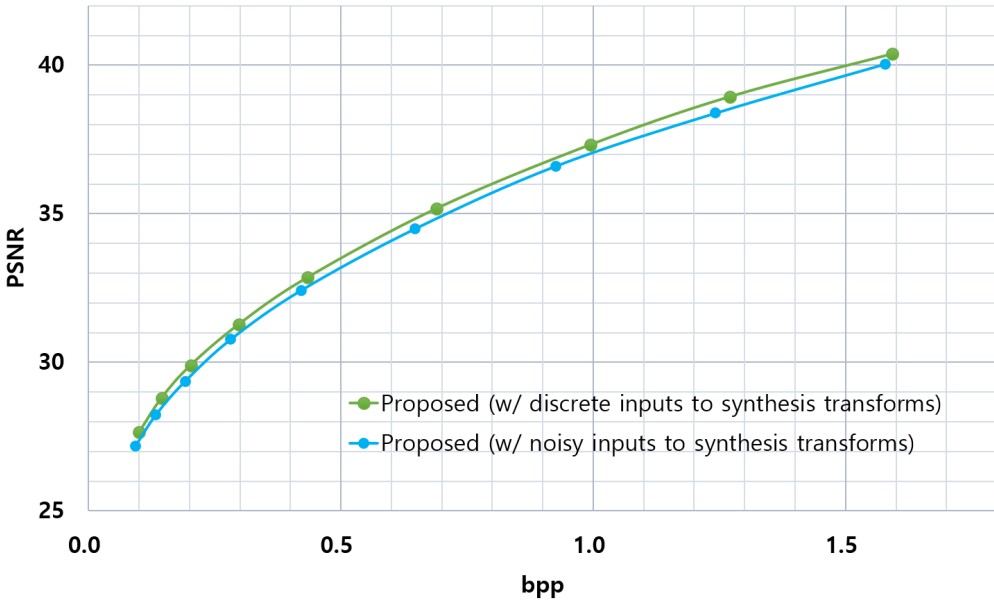

Figure 8: Evaluation results of the models trained using noisy/discrete representations as input to synthesis transforms

## 6.3 SAMPLES OF THE EXPERIMENTS

In this section, we provide a few more supplemental test results. Figures 9, 10, and 11 show the results of the MSE optimized version of our method, whereas figures 12 and 13 show the results of the MS-SSIM optimized version. All figures include a ground truth image and the reconstructed images for our method, BPG, and Ballé et al. (2018)'s approach, in the clockwise direction.

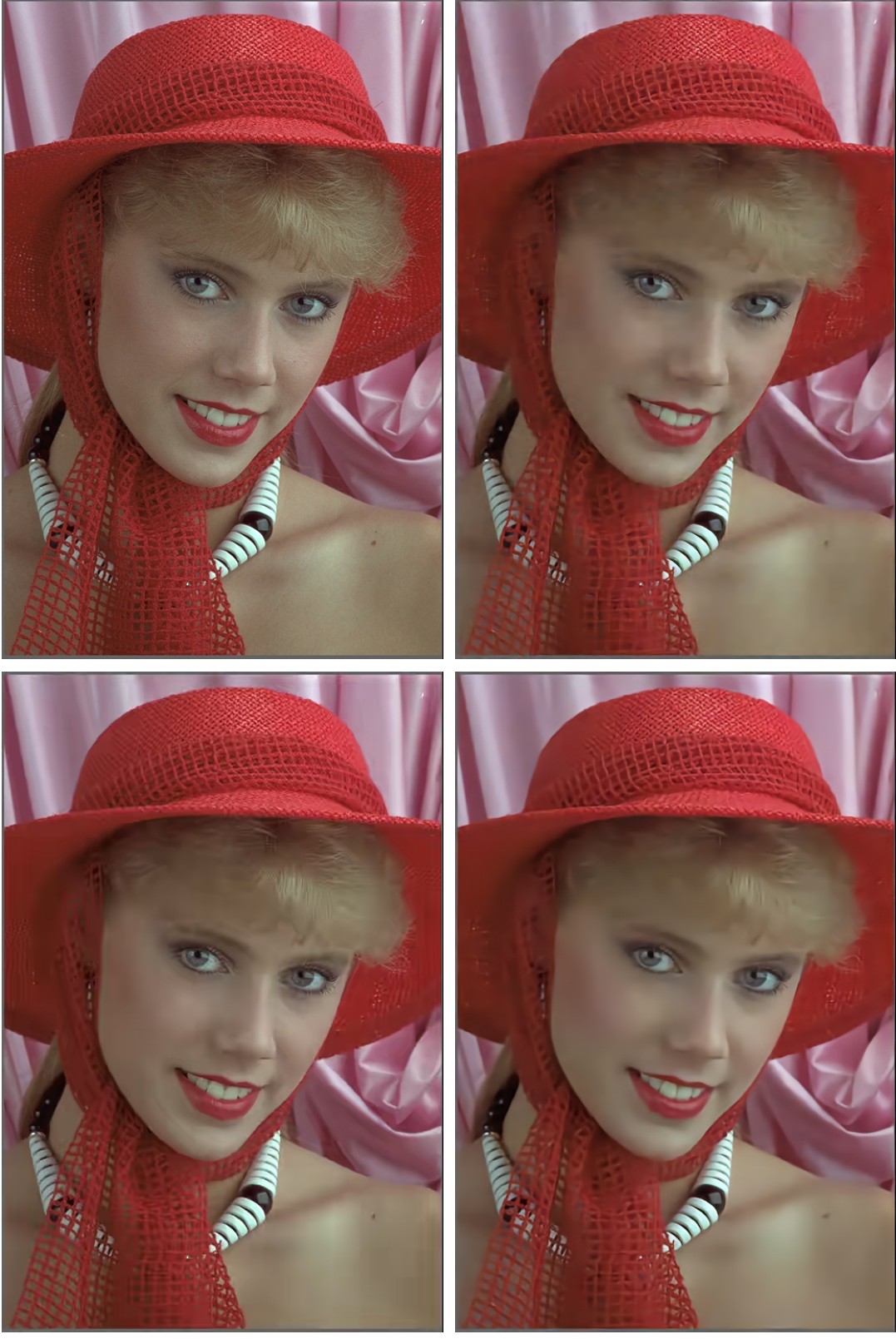

Figure 9: Sample test results. Top left, ground truth; top right, our method (MSE optimized; bpp, 0.2040; PSNR, 32.2063); bottom left, BPG (bpp, 0.2078; PSNR, 32.0406); bottom right, Ballé et al. (2018)'s approach (MSE optimized; bpp, 0.2101; PSNR, 31.7054)

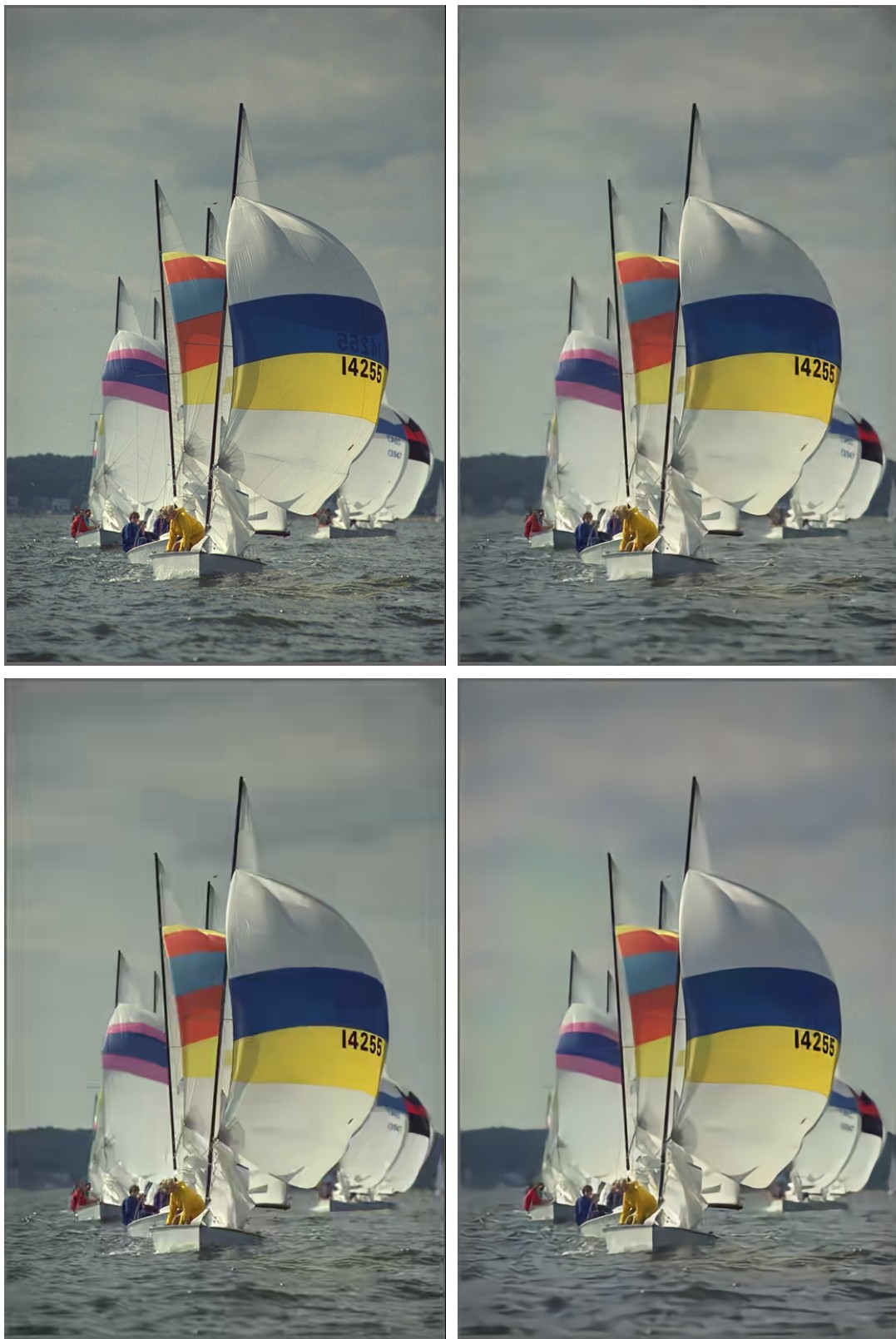

Figure 10: Sample test results. Top left, ground truth; top right, our method (MSE optimized; bpp, 0.1236; PSNR, 32.4284); bottom left, BPG (bpp, 0.1285; PSNR, 32.0444); bottom right, Ballé et al. (2018)'s approach (MSE optimized, bpp, 0.1229; PSNR, 31.0596)

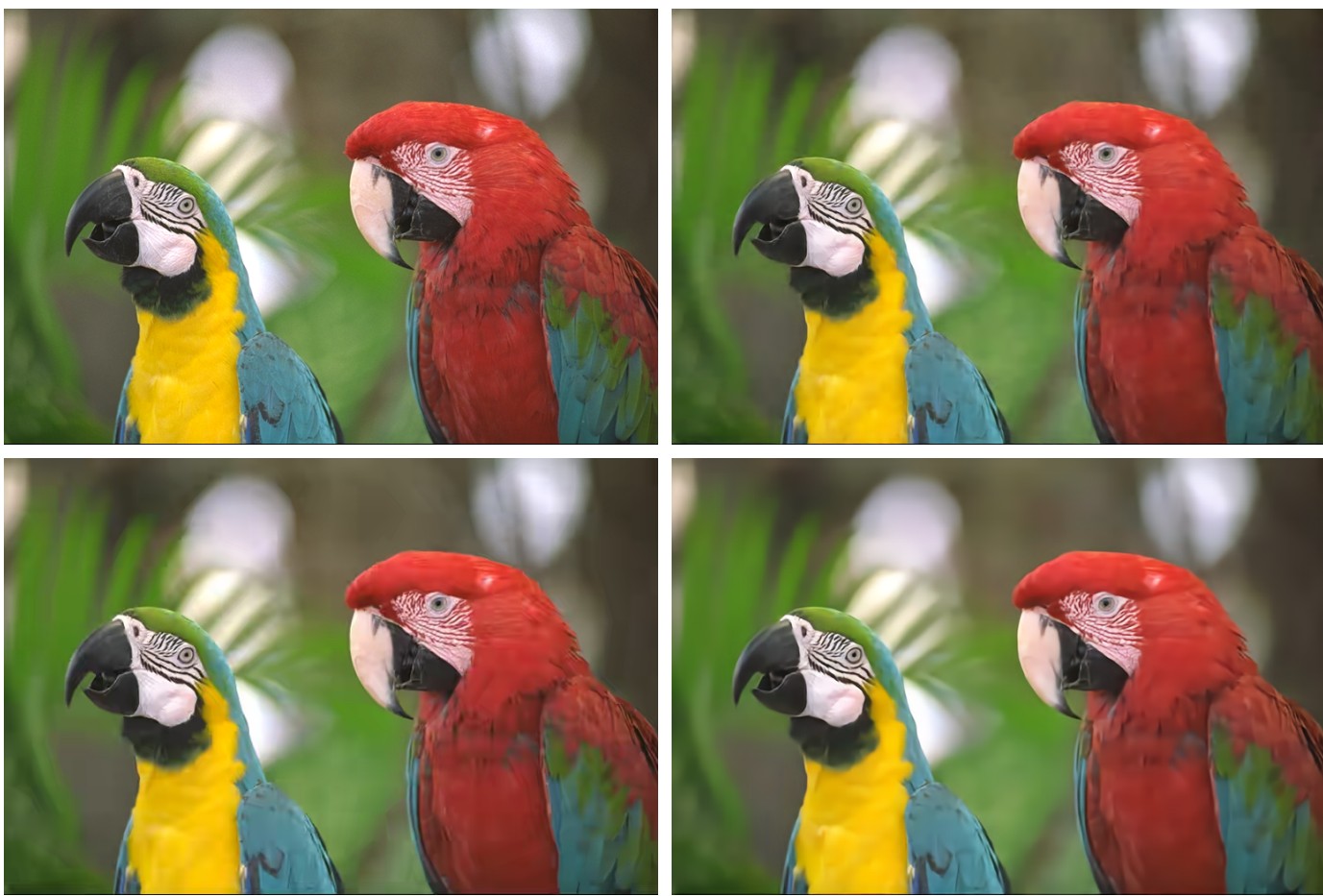

Figure 11: Sample test results. Top left, ground truth; top right, our method (MSE optimized; bpp, 0.1501; PSNR, 34.7103); bottom left, BPG (bpp, 0.1477; PSNR, 33.9623); bottom right, Ballé et al. (2018)'s approach (MSE optimized; bpp, 0.1520; PSNR, 34.0465)

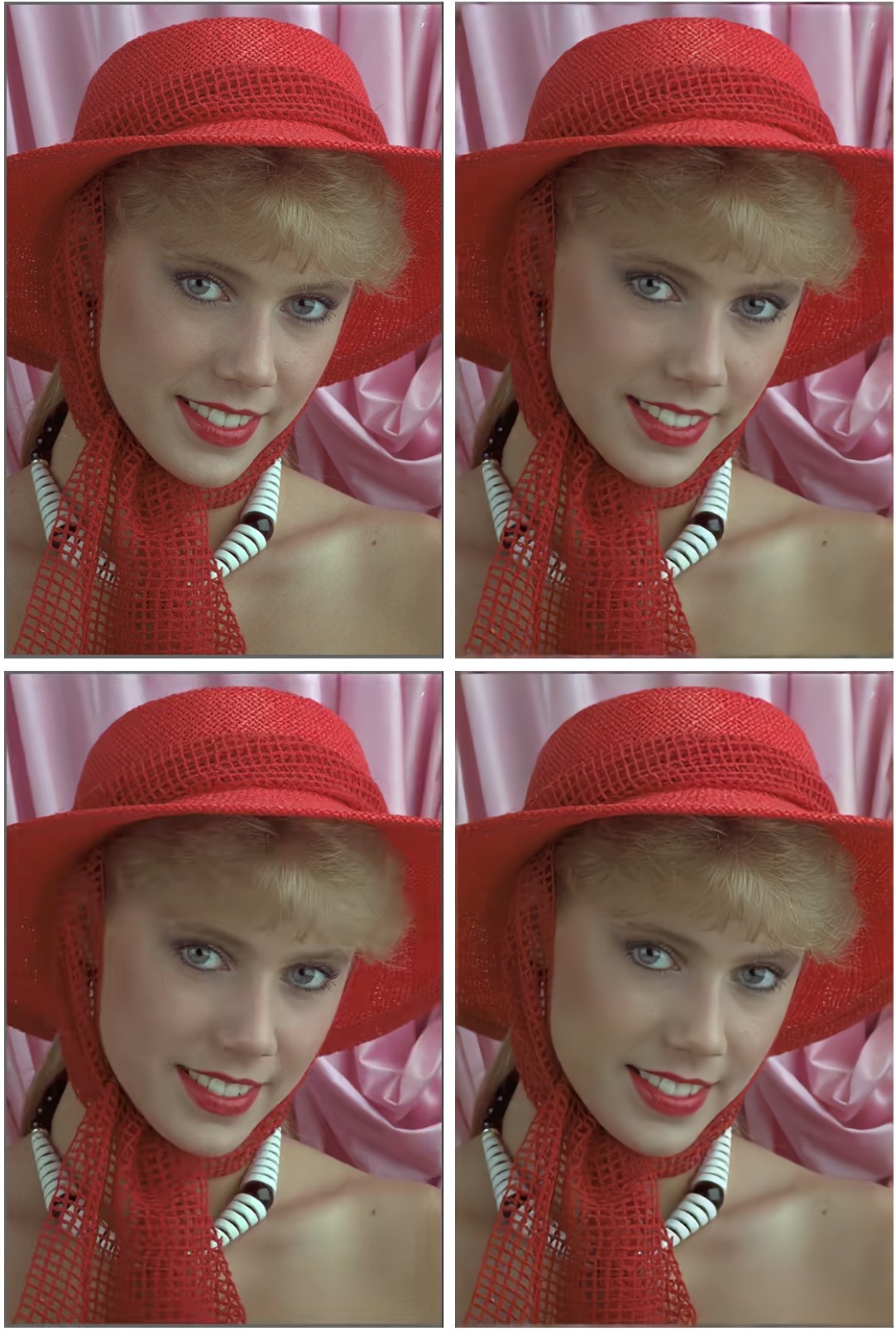

Figure 12: Sample test results. Top left, ground truth; top right, our method (MS-SSIM optimized; bpp, 0.2507; MS-SSIM, 0.9740); bottom left, BPG (bpp, 0.2441; MS-SSIM, 0.9555); bottom right, Ballé et al. (2018)'s approach (MS-SSIM optimized; bpp, 0.2101; MS-SSIM, 0.9705)

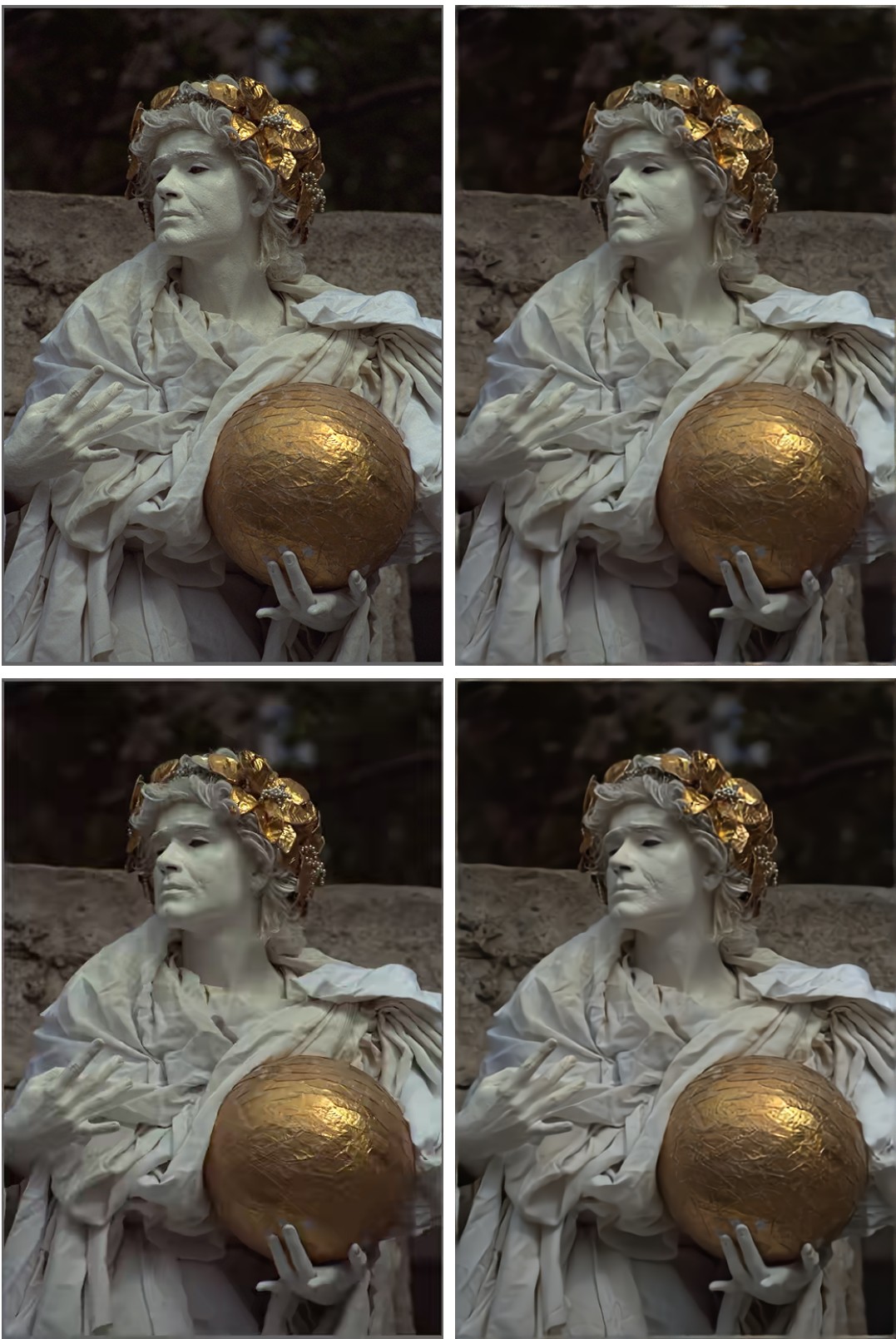

Figure 13: Sample test results. Top left, ground truth; top right, our method (MS-SSIM optimized; bpp, 0.2269; MS-SSIM, 0.9810); bottom left, BPG (bpp, 0.2316; MS-SSIM, 0.9658); bottom right, Ballé et al. (2018)'s approach (MS-SSIM optimized; bpp, 0.2291; MS-SSIM, 0.9786)

