# OpenReview forum: "Context-adaptive Entropy Model for End-to-end Optimized Image Compression"
_ICLR.cc/2019/Conference_

### Official Review · AnonReviewer3 · 2018-10-26
**A New Context-adaptive Entropy Model for Image Deep Compression**

**Rating:** 6
**Confidence:** 3

**Review:**

Summary. The paper is an improvement over (Balle et al 2018) for end-to-end image compression using deep neural networks. It relies on a generalized entropy model and some modifications in the training algorithm. Experimentals results on the Kodak PhotoCD dataset show improvements over the BPG format in terms of the peak signal-to-noise ratio (PSNR). It is not said whether the code will be made available.

Pros.
* Deep image compression is an active field of research of interest for ICLR. The paper is a step forward w.r.t. (Balle et al 2018).
* The paper is well written.
* Experimental results are promising.

Cons.
* Differences with (Balle et al 2018) should be emphasized. It is not easy to see where the improvements come from: from the new entropy model or from modifications in the training phase (using discrete representations on the conditions).
* I am surprised that there is no discussion on the choice of the hyperparameter \lambda: what are the optimal values in the experiments? Are the results varying a lot depending on the choice? Is there a strategy for an a priori choice?
* Also is one dataset enough to draw conclusions on the proposed method?

Evaluation.
As a non expert in deep learning compression, I have a positive opinion on the paper but the paper seems more a fine tuning of the method of (Balle et al 2018). Therefore I am not convinced that the improvements are sufficiently innovative for publication at ICLR despite the promising experimental results.

Some details.
Typos: the p1, the their p2 and p10, while whereas p3, and and figure 2
p8: lower configurations, higher configurations, R-D configurations

---

> ### Author Response · Authors · 2018-11-16
> **Response to AnonReviewer3's comments**
>
> Dear reviewer 3,
>
> [Authors] First of all, we really appreciate your careful comments. Please understand our late response due to an additional experiments to resolve your concerns. Attached please find the revised version. We address your comments as below:
>
> ______
> Cons.
> * Differences with (Balle et al 2018) should be emphasized. It is not easy to see where the improvements come from: from the new entropy model or from modifications in the training phase (using discrete representations on the conditions).
> ______
> [Authors] We agree with your comments and we also think it is a really important point that needs to be clarified. To clarify this, we conducted an additional experiments (appendix 6.2 in the revised version) on the network trained using the noisy representations as inputs of g_s and h_s. From the results, we found that the performance improvement comes from both new context adaptive entropy model and replacing the noisy representations with the discrete representations. Compared with (Balle et al 2018)’s approach, our network trained with the noisy representations is 11.97% better in compression performance, whereas the same trained with the discrete representations is 7.2% better.
> ______
>
> * I am surprised that there is no discussion on the choice of the hyperparameter \lambda: what are the optimal values in the experiments? Are the results varying a lot depending on the choice? Is there a strategy for an a priori choice?
> ______
> [Authors] As your comments, \lambda is very important parameter for training, which determines which to focus on between rate and distortion. However, \lambda is not an optimization target, but a given condition for optimization. Therefore, several networks were trained, each of which was trained with a specific value of \lambda. In figure 5, illustrating the evaluation results, each point represents a result of one single network trained under a specific \lambda, so one line of our approach represents results of nine trained networks. We described the range of \lambda values in section 4.2, from 0.01 to 0.5. As the lambda increases, the gain of the bit amount side is increased, but the loss of the image quality side is also increased. The exact values that we used are 0.5, 0.4, 0.3, 0.2, 0.1, 0.06, 0.03, 0.017, and 0.01, in order of from rate-centric condition to distortion-centric condition.
> To clarify the purpose of using \lambda, we have added more description about \lambda in the revised version, before equation (2).
> ______
>
> * Also is one dataset enough to draw conclusions on the proposed method?
> ______
> [Authors] One dataset may not be enough for completely evaluating one method. However, Kodak photo CD image set has served as a reference test set for many studies. We guess that the reason many studies have used this set is to make comparison between approaches easier, and to ensure the objectivity of the comparison results. Instead of adding more evaluation results over other image sets, we will add an URL link to our test code repository if publication is decided. Our methods could be evaluated over any kind of image sets with the test code.
> ______
>
> Evaluation.
> As a non expert in deep learning compression, I have a positive opinion on the paper but the paper seems more a fine tuning of the method of (Balle et al 2018). Therefore I am not convinced that the improvements are sufficiently innovative for publication at ICLR despite the promising experimental results.
> ___
> [Authors] (Balle et al 2018) successfully captures spatial dependencies of natural images by estimating scales of representation, in an input-adaptive manner. To further remove the spatial dependency, we proposed a model that can sequentially predict each value (mean) of representations, as well as standard deviation values as in (Balle et al 2018). We believe that this autoregression using the two types of contexts is essential component to achieve higher compression performance.
> It has been just two years since two great papers, which become a basis of entropy model based image compression, were poposed by (Balle et al. 2017) and (Theis et al. 2017), and currently context utilization within latent space is at the very beginning phase. We believe that a variety of context utilization methods will be studied, and hope our work will serve as a stepping stone for future studies utilizing various types of bit-free and bit-consuming contexts.
> ______
>
> Some details.
> Typos: the p1, the their p2 and p10, while whereas p3, and and figure 2
> [Authors] We’ve fixed the typos. Thank you for pointing out.
> p8: lower configurations, higher configurations, R-D configurations
> [Authors] We’ve changed the phrases to make them clear, as follows:
> \lambda configurations for lower bit-rates, \lambda configurations for higher bit-rates, \lambda configurations
>
>
> Thank you very much for your insightful comments again!
>
> Regards,
> Authors

---

### Official Review · AnonReviewer1 · 2018-11-02
**not enough value added**

**Rating:** 7
**Confidence:** 5

**Review:**

Update:
I have updated my review to mention that we should accept this work as being concurrent with the two papers that are discussed below.

Original review:
This paper is very similar to two previously published papers (as pointed by David Minnen before the review period was opened):
"Learning a Code-Space Predictor by Exploiting Intra-Image-Dependencies" (Klopp et al.)  from BMVC 2018,
and
"Joint Autoregressive and Hierarchical Priors for Learned Image Compression" (Minnen et al.) from NIPS 2018.

The authors have already tried to address these similarities and have provided a list in their reply, and my summary of the differences is as follows (dear authors: please comment if I am misrepresenting what you said):
(1) the context model is slightly different
(2) parametric model for hyperprior vs non-parametric
(3) this point is highly debatable to be considered as a difference because the distinction between using noisy outputs vs quantized outputs is a very tiny detail (any any practitioner would probably try both and test which works better).
(4) this is not really a difference. The fact that you provide details about the method should be a default! I want all the papers I read to have enough details to be able to implement them.
(5+)  not relevant for the discussion here.

If the results were significantly different from previous work, these differences would indeed be interesting to discuss, but they didn't seem to change much vs. previously published work.

If the other papers didn't exist, this would be an excellent paper on its own. However, I think the overlap is definitely there and as you can see from the summary above, it's not really clear to me whether this should be an ICLR paper or not. I am on the fence because I would expect more from a paper to be accepted to this venue (i.e., more than an incremental update to an existing set of models, which have already been covered in two papers).

---

> ### Author Response · Authors · 2018-11-16
> **Response to AnonReviewer1's comments**
>
> Dear reviewer 1,
> We appreciate your comments. As discussed in the separate thread, the most important issue is to clarify the criteria on prior works. To deal with this issue, we officially requested the decision of AC/PCs, and currently we’re waiting for it.
>
> We agree with most of your comments, but please understand that the reason we described the differences was to emphasize that our work was independently conducted. One more thing we’d like to emphasize is that our results were significantly superior from Klopp et al. (2018)'s approach. As we described in other postings, our work is more than 10% superior in compression performance.
>
> As we have described in the response to reviewer 2’s comments, occasionally there exist concurrently conducted studies, such as DiscoGAN and CycleGAN. Although decision on prior works will be made by AC/PCs, we would also be grateful if you view our work from a generous perspective for mutual progress of technologies.
>
> Regards,
> authors

---

### Official Review · AnonReviewer2 · 2018-11-05
**Minor improvements over existing work**

**Rating:** 7
**Confidence:** 4

**Review:**

The authors present their own take on a variational image compression model based on Ballé et al. (2018), with some  interesting extensions/modifications:

- The combination of an autoregressive and a hierarchical approach to define the prior, as in Klopp et al. (2018) and Minnen et al. (2018).
- A simplified hyperprior, replacing the flow-based density model with a simpler Gaussian.
- Breaking the strict separation of stochastic variables (denoted with a tilde, and used during training) and deterministic variables (denoted with a hat, used during evaluation), and instead conditioning some of the distributions on the quantized variables directly during training, in an effort to reduce potential training biases.

The paper is written in clear language, and generally well presented with a great attention to detail. It is unfortunate that, as noted in the comments above, two prior, peer-reviewed studies have already explored extensions of the prior by introducing an autoregressive component, obtaining similar results.

As far as I can see, this reduces the novelty of the present paper to the latter two modifications. The bit-free vs. bit-consuming terminology is simply another way of presenting the same concept. In my opinion, it is not sufficiently novel to consider acceptance of this work into the paper track at ICLR.

The authors should consider to build on their work further and consider publication at a later time, possibly highlighting the latter modifications. However, the paper would need to be rewritten with a different set of claims.

Update: Incorporating the AC/PC decision to treat the paper as concurrent work.

---

> ### Author Response · Authors · 2018-11-16
> **Response to AnonReviewer2's comments**
>
> Dear reviewer 2,
>
> We appreciate your comments. As discussed in the separate thread, the most important issue is to clarify the criteria on prior works. To deal with this issue, we officially requested the decision of AC/PCs, and currently we’re waiting for it.
>
> Although the current prior work issue depends on the chairs’ decision, we’d like to show one similar example, the case of DiscoGAN (https://arxiv.org/abs/1703.05192) and CycleGAN (https://arxiv.org/abs/1703.10593). Both were opened to public via arXiv March 2017 (15 days of time difference). In spite of their very similar concepts and structures, they were accepted by ICML2017 and ICCV2017, respectively.
>
> In addition, from the technical point of view, our approach has clear difference from Klopp et al. (2018)'s approach in performance (our approach is more than 10% superior in compression performance) and paper composition (we provide more comprehensive and concrete models along with a detailed implementation and training methods.) We think that around 10% of performance improvement has value enough to be reported to public in the compression research field.
>
> Regards,
> authors

---

### Public Comment · ~David_Minnen1 · 2018-10-05
**quality research but (very recent) papers present a similar model**

"Learning a Code-Space Predictor by Exploiting Intra-Image-Dependencies" (Klopp et al.) was recently published at BMVC 2018 (http://bmvc2018.org/contents/papers/0491.pdf) and also explores the use of spatial context to improve rate-distortion (RD) performance for learned image compression. In addition to using context to predict the parameters of the entropy model, they introduce a new nonlinearity (a sparse variant of GDN) and generalize the entropy model by using an equal-weight mixture of Gaussians.

"Joint Autoregressive and Hierarchical Priors for Learned Image Compression" (Minnen et al.) was accepted at NIPS 2018 (https://arxiv.org/abs/1809.02736) and presents a very similar model. This paper combines information from the hyperprior (which I think is the same as the "bit-consuming context" in the submitted ICLR paper) and an autoregressive model ("bit-free context") in a slightly different way than Klopp et al. resulting in improved RD performance.

The submitted ICLR paper shows RD curves that are very similar to those in Minnen et al. and better than those in Klopp et al. It also introduces the idea of splitting the latent representation into two parts and coding each part with a different entropy model. This split makes sense since many latent values are zero and thus may not benefit from context or a predicted mean for the Gaussian entropy model. I agree with the authors' claim that this split should reduce runtime, though I'm not sure how significant it will be relative to the total encode / decode time (some runtime data would help here, though neither of the papers cited above provide runtime data so I don't think it's a requirement for research focused on improving RD performance).

In my opinion, the quality of the research and results presented in the submitted paper are appropriate for publication at ICLR. However, the model is too similar to Klopp et al. and Minnen et al. and thus should not be accepted without further differentiation.

---

> ### Author Response · Authors · 2018-10-11
> **Response to comments (Cont'd)**
>
> (Cont'd)
>
> 3) Balle et al. (2018)’s approach (https://arxiv.org/abs/1802.01436) uses noisy representations y_tilde and z_tilde for training to deal with the discontinuities caused by quantization for the entire model, including inputs to the transforms, g_s (Decoder) and h_s (Hyper Decoder) as well as for inputs to entropy model functions (continuous model functions convolved with an uniform dist. function). Minnen et al.’s approach seems to deal with the discontinuities in the same manner as in Balle et al. (2018)’s approach. Therefore, we think that model expressions in Minnen et al.’s paper represent the target model for test, because they use the “hat” symbols not only for conditions, but also for inputs. It seems that all these quantized representations are replaced by noisy representations, for training, as Balle et al. (2018) did. On the other hand, as clearly noted in our manuscript, we only use noisy representations as inputs to the entropy model functions, whereas quantized representations are used as inputs to the transforms for training. We made such a model because it prevents mismatches between training and testing and provides better performance. If necessary, we will provide the quantitative results on the impact of the two types of transform inputs for training. The representation flows for training, for our work and Minnen et al.’s approach, are different from each other as below:
>
>   * Our approach (when training):
>     x -> [g_a;Encoder] -> y -> [q] -> y_hat -> [g_s;Decoder] -> x’
>     y_hat -> [h_a;Hyper Encoder] -> z -> [q] -> z_hat -> [h_s; Hyper Decoder] -> C’;Psi
>
>   * Minnen et al.’s approach (when training):
>     x -> [g_a;Encoder] -> y -> [U] -> y_tilde -> [g_s;Decoder] -> x’
>     y -> [h_a;Hyper Encoder] -> z -> [U] -> z_tilde -> [h_s; Hyper Decoder] -> C’;Psi
>
> Note that we used quantized y_hat as inputs to h_a, but it has nothing to do with the mismatches between training and testing. We used them to match inputs of h_a to target representations of model estimation.
>
> 4) In our paper, we provided details for training and implementation of our proposed model. For example, we provide information on training sets, batch sizes, the number of training iterations, optimization algorithms, learning rates, which are not given in Minnen et al.’s paper, and also techniques for reducing large training costs, such as random index selection, so that the readers can implement and train their own models without much trial and error. Furthermore, we are planning to share our test code via github after code refactoring and exception handling. We hope that our work and test code will draw more interest in the ANN-based image compression field.
>
> 5) We presented the directions of improvement from a different perspective. Current ANN-based image compression techniques are still not practical due to high complexity versus low gain. To solve this, we need to maximize compression performance or reduce complexity. At the end of the paper by Minnen et al., they presented one direction to obtain fast, low-complexity solutions, while we suggested the use of high-level contexts to maximize compression performance. Both directions would be important topics for various follow-up studies.
>
> 6) As the commenter's suggestion, we will add the runtime data of our hybrid entropy model. The hybrid model can be viewed as one implementation technique.
>
> 7) In addition, we are now aware that our results are not the first results that outperform BPG in terms of PSNR, so we will remove the related phrases and sentences.
>
> We hope that our work will be a solid evidence supporting Minnen et al.’s work, and hope both will together suggest a promising direction for ANN-based image compression researches.

---

> ### Author Response · Authors · 2018-10-11
> **Response to comments**
>
> We appreciate your comments. Last month, we did not notice that two papers were open to public because we were concentrating on editing work of our manuscript. Thank you for introducing the two excellent papers. I am surprised and also pleased with the fact that similar ideas for entropy models have been already proposed. We have read the two papers carefully, and the followings are our response to comments:
>
> In the case of Klopp et al.'s paper (http://bmvc2018.org/contents/papers/0491.pdf), they focused on improving GDN and predicting distributions of latent variables using given surrounding variables, and they achieved noticeable results on those topics. In addition, their supplementary material (http://bmvc2018.org/contents/supplementary/pdf/0491_supp.pdf) includes an integrated model with a similar structure to that of our method, along with a simple description (section 8.2.3 and figure 8 of the supplementary material). Experimental results of the integrated model, in terms of the MS-SSIM, are represented in the main text of the paper. However, as the experimental results show, the degree of performance improvement over the existing approaches decreases as the bpp increases. Our approach (and Minnen et al.’s approach) shows that as bpp increases, the performance gain increases, which shows that our approach takes full advantage of the hierarchical prior (or hyperprior). The paper by Klopp et al. only introduced basic mechanism of integration using two contexts, but did not describe the details of the integration method. On the other hand, in the scope of the integration, we provide a concrete model that fully utilizes two contexts, a detailed implementation method, and verification results of the proposed model through performance comparisons.
>
> Regarding Minnen et al.’s approach (https://arxiv.org/abs/1809.02736), we agree that the goals, structures, and results of their method are very similar to ours. It is an hornor to us that we have privilege of comparing our work with the excellent paper adopted for NIPS. We hope to publish our paper in ICLR to support Minnen et al.’s work and thereby help presenting one promising direction of research for ANN-based image compression. The differences between Minnen et al.’s approach and our work, which could be considered, are as follows:
>
> 1) One of the most important parts of our work is to propose a model that can predict probability distributions of latent representation based on two types of contexts, context consuming bits and context not using it. From this point of view, our model clearly distinguishes between context extraction and distribution prediction, and this allows our model to provide extensibility. For example, if we want to use multi-scale context information, which is one of our candidates for further work, we only need to replace the extraction model with new one. On the other hand, Context Model of Minnen et al.’s approach includes both the context extraction and the transform function. Regarding the hyperpriors, the compositions of the Hyper Encoder / Decoder and the utilization of the results are also very similar to each other, but in our work, the information obtained through the h_a (Hyper Encoder) / h_s (Hyper Decoder) is strictly defined as one type of the contexts, and this allows our model to be a framework that can accommodate various contexts in the future.
>
> 2) We used a parametric model (Gaussian dist.) for an entropy model of hyperprior z, whereas the non-parametric model is used in Minnen et al.’s approach. There are advantages and disadvantages to both non-parametric and parametric models. However, if there is no significant difference in performance between the two cases, we think that the advantage of the parametric model, which is easy to implement and cost-efficient, can be highlighted. As stated by the commenter, our approach and Minnen et al.’s approach show very similar performance results, which demonstrates that our simple parametric model provides sufficient performance for modeling the hyperprior z.

---

### Author Response · Authors · 2018-11-06
**Our opinion on the criteria of prior works**

Dear reviewers, area chair, and program chairs,

First of all, thank you for the careful reviews and for giving this rebuttal chance. Before replying to each review comment, we would like to give our opinion on two reviewers’ comments regarding prior works.

Reviewer 1 and 2 posted comments that there exist two similar prior works as follows:
-	Klopp et al.'s paper (http://bmvc2018.org/contents/papers/0491.pdf), published on September 3rd
-	Minnen et al.’s paper (https://arxiv.org/abs/1809.02736), uploaded to arXiv on September 8th

However, in our opinion, our work should be recognized as a concurrent and independent work, because related standards are documented in the ICLR Reviewer Guidelines (https://iclr.cc/Conferences/2019/Reviewer_Guidelines), leaving out a long period of time for designing, testing and validating works, which took more than half a year.

Specifically, according to the ICLR reviewer guidelines, it is clearly stated that the papers opened less than 30 days prior to the ICLR deadline should not be considered as prior works, as follows:

-	"While we encourage reviewers to apply the reasonable standards of the relevant community in considering what does and does constitute prior work, the following minimum standards will be enforced: no paper will be considered prior work if it appeared on arxiv, or another online venue, less than 30 days prior to the ICLR deadline."

Klopp et al.'s paper was published on September 3rd, and Minnen et al.'s paper was uploaded to arXiv on September 8th. Because both appeared less than 30 days prior to the ICLR deadline, they cannot be considered as prior works in ICLR’s review process.

As noted in the detailed reviews given by reviewer 1 and 2, the main problem on our paper they pointed out is existence of prior works, and we believe that this cause low scores from the reviewers. We respect the comments from the reviewers and the two papers, but we think that the "minimum standard” on the prior works should be a rule for all papers submitted into ICLR. Therefore, we believe that our paper should be revaluated without considering the two papers as prior works.

Numerous studies are being conducted on the same subject at the same time, and there may be occasional ambiguities in precedence relation. We believed that ICLR, as one of the top conferences, has achieved fairness by providing a clear review basis on these issues. We hope that our paper will not become one exceptional case. We would deeply appreciate it if you get the related discussion started.

---

> ### Comment · AnonReviewer2 · 2018-11-06
> **Committed to resolve this issue fairly**
>
> Thank you for alerting us to this section of the reviewer guidelines.
>
> First of all, there was no intention to say that the paper was plagiarizing other work in any way. I believe that the authors did good research, the clarity and thoroughness of the paper demonstrates this.
>
> However, Klopp's work had been peer-reviewed and published prior to the deadline (the conference date was 3 September). I believe the section in the reviewer guidelines you pointed out (in particular, the 30-day grace period) is specifically applied to arXiv and other pre-print sites which do not include peer review.
>
> So, while you are right that Minnen's work should be disregarded (because the peer reviewed paper isn't yet available), Klopp's paper should still count as prior work, as it had been peer reviewed well before the ICLR deadline.
>
> Ultimately, I think it would be best for the area/program chair to decide this. In case they decide Klopp's paper should count as concurrent work, I would be in favor of accepting the paper, as it is a high quality paper besides the novelty aspect.

---

> > ### Author Response · Authors · 2018-11-07
> > **Response to AnonReviewer2's comments**
> >
> > First of all, we really appreciate your quick reply. We believe that there’s an ICLR’s standard on the scope of the reviewer's guideline. In any case, we would appreciate it if you recognize that our work was conducted independently.
> >
> > In addition, as mentioned in our previous posting (https://openreview.net/forum?id=HyxKIiAqYQ&noteId=SyxV-q_3cm), there are obvious differences between ours and Klopp et al.’s approach, in the following aspects:
> >
> >   -	In terms of performance, there is a significant difference between ours and the Klopp et al.’s approach. In the Klopp et al.’s paper, comparison results are provided, only in terms of the MS-SSIM. Comparing the experimental results in the same environment (MS-SSIM over the Kodak set), our method is more than 10% superior (Ours: -13.93%, Klopp et al: -3.2%; when both are compared with Balle et al. (2018)’s approach).
> >
> >   -	In terms of paper composition, we provide a concrete model that fully utilizes two contexts, a detailed implementation method, and verification results of the proposed model, whereas only basic mechanism of integrating the two contexts are provided by Klopp et al. (because they mainly focused on other points such as improving GDN and predicting distributions of latent variables using given surrounding variables.)
> >
> > Therefore, we think that our approach has enough value as an academic paper for readers and subsequent studies.

---

> > > ### Comment · AnonReviewer2 · 2018-11-22
> > > **Thanks for clarifications**
> > >
> > > Thank you for pointing these things out.
> > >
> > > I also feel sympathetic towards AnonReviewer1's comments. However, in practice, we have to draw the line somewhere. Where this line is should ultimately be decided by the conference organizers.
> > >
> > > I think all the relevant information is now on the table. Let's hope that the AC/PC can provide some guidance.

---

> ### Comment · AnonReviewer1 · 2018-11-06
> **decision w.r.t. this paper should be done by the AC/PC**
>
> I agree with AnonReviewer2 that this is a complicated issue. From a technical standpoint, this is a good paper, but we have the issue that was outlined multiple times w.r.t. considering this either "concurrent work" or not accepting it. If we are to consider it concurrent work, I wouldn't oppose this decision.
>
> I do agree that it's very unfortunate from a timing standpoint that most of the conversation is not focused on the technical aspects, which seem OK, but rather the issue of how to handle this paper, but sometimes good ideas come from multiple people roughly at the same time.
>
> There's a question that we (the academic field) must ask ourselves and decide what is the correct course of action, and perhaps even provide guidance to reviewers with respect to this: what is the correct way to handle a situation like this?
>
> The paragraph cited by the authors provides some guidance, but I don't think it's sufficient. I don't think the work itself could have been plagiarized in any way from Klopp et al given the time frame, so we don't have to worry about that aspect. The work itself is non-trivial, so we can definitely expect that it took a considerable amount of time (several months), yet it ended up being quite similar. Do we (academics) want to reward this work regardless of the similarities by accepting this paper? If yes, I would really like to see this being clarified in the review policy.
>
> On one hand if we accept this, future researchers will need to figure out which of the three papers to cite when referring to this type of idea (it's unlikely that all three papers will be cited). On the other hand, if we don't accept it, but we agree that the idea was developed independently, then are we doing the authors a disservice?
>
> I am very frustrated because the answer to any of the questions above is not at all clear to me.
>
> I would really appreciate if the AC/PC provided some insight here.

---

> > ### Author Response · Authors · 2018-11-07
> > **Response to AnonReviewer1's comments**
> >
> > We deeply appreciate your reply. We agree with your insightful comments and concerns. Regarding your concerns, one more thing we would like to add is as follows:
> >
> > Image coding using the context of latent space is now at the beginning phase, and various subsequent studies are expected to proceed. In these following studies, citing multiple papers may be a burden more or less. However, the three studies differ in perspectives on contexts, implementation details, training methods, and directions for future studies, so these differences would rather be a chance to provide richer technical evidences and insights for them. Since each of the three papers has its own pros and cons, we think that citation will be naturally decided by future studies.
> >
> > Please refer to our previous postings for detailed differences between the papers.
> > - https://openreview.net/forum?id=HyxKIiAqYQ&noteId=SyxV-q_3cm
> > - https://openreview.net/forum?id=HyxKIiAqYQ&noteId=SJxyx5u397

---

### Meta-Review · Area_Chair1 · 2018-12-14
**Strong contribution to image compression**

**Confidence:** 5
**Recommendation:** Accept (Poster)

**Metareview:**

This paper proposes an algorithm for end-to-end image compression outperforming previously proposed ANN-based techniques and typical image compression standards like JPEG.

Strengths
- All reviewers agreed that this a well written paper, with careful analysis and results.

Weaknesses
- One of the points raised during the review process was that 2 very recent publications propose very similar algorithms. Since these works appeared very close to ICLR paper submission deadline (within 30 days), the program committee decided to treat this as concurrent work.

The authors also clarified the differences and similarities with prior work, and included additional experiments to clarify some of the concerns raised during the review process. Overall the paper is a solid contribution towards improving image compression, and is therefore recommended to be accepted.